# Extracellular Vesicles, Circadian Rhythms, and Cancer: A Comprehensive Review with Emphasis on Hepatocellular Carcinoma

**DOI:** 10.3390/cancers16142552

**Published:** 2024-07-16

**Authors:** Baharan Fekry, Lierni Ugartemendia, Nestor F. Esnaola, Laura Goetzl

**Affiliations:** 1McGovern Medical School, University of Texas Health Science Center at Houston, Houston, TX 77030, USA; lierni.ugartemendiaugalde@uth.tmc.edu (L.U.); laura.goetzl@uth.tmc.edu (L.G.); 2Division of Surgical Oncology and Gastrointestinal Surgery, Department of Surgery, Houston Methodist Hospital, Houston, TX 77030, USA; nfesnaola@houstonmethodist.org

**Keywords:** hepatocellular carcinoma, circadian rhythms, cancer biomarkers, extracellular vesicles, exosomes, exosomal cargo, early cancer detection

## Abstract

**Simple Summary:**

Extracellular vesicles (ECVs), especially exosomes, play a crucial role in hepatocellular carcinoma (HCC) by facilitating intercellular communication and influencing circadian rhythms, thereby affecting HCC progression and treatment responses. Exosomes serve as biomarkers for early cancer detection, drug delivery vehicles, and modulators of immune responses. They promote angiogenesis, modulate the tumor microenvironment, and spread drug resistance. Advanced techniques for isolating cell-specific exosomes and aligning treatments with natural circadian rhythms show promise for early detection and personalized therapies. Understanding how exosomal cargo is sorted and interacts with circadian genes could revolutionize HCC diagnostics and treatments, improving patient outcomes. Utilizing liver-specific proteins for precise exosome isolation can enhance early detection and treatment efficacy, paving the way for more personalized and effective cancer therapies.

**Abstract:**

This review comprehensively explores the complex interplay between extracellular vesicles (ECVs)/exosomes and circadian rhythms, with a focus on the role of this interaction in hepatocellular carcinoma (HCC). Exosomes are nanovesicles derived from cells that facilitate intercellular communication by transporting bioactive molecules such as proteins, lipids, and RNA/DNA species. ECVs are implicated in a range of diseases, where they play crucial roles in signaling between cells and their surrounding environment. In the setting of cancer, ECVs are known to influence cancer initiation and progression. The scope of this review extends to all cancer types, synthesizing existing knowledge on the various roles of ECVs. A unique aspect of this review is the emphasis on the circadian-controlled release and composition of exosomes, highlighting their potential as biomarkers for early cancer detection and monitoring metastasis. We also discuss how circadian rhythms affect multiple cancer-related pathways, proposing that disruptions in the circadian clock can alter tumor development and treatment response. Additionally, this review delves into the influence of circadian clock components on ECV biogenesis and their impact on reshaping the tumor microenvironment, a key component driving HCC progression. Finally, we address the potential clinical applications of ECVs, particularly their use as diagnostic tools and drug delivery vehicles, while considering the challenges associated with clinical implementation.

## 1. Introduction

Extracellular vesicles (ECVs), including exosomes, are secreted by diverse cell types, ranging from fibroblasts to cancer cells, and play a critical role in facilitating the intercellular transmission of signals through bioactive molecules like nucleic acids, proteins, lipids, and metabolites. ECVs can vary in size, content, and surface markers, reflecting the diverse origins of their parent cells. This heterogeneity influences extracellular vesicle (ECV) function, as different subpopulations may have distinct effects on recipient cells [1,2,3]. The significance of ECVs is underscored by their ubiquitous presence in diverse body fluids, including bile, blood, breast milk, urine, cerebrospinal fluid, and saliva [4,5,6].

The circadian rhythm functions as an internal biological clock, regulating physiological activities throughout a 24 h period. The circadian rhythm’s impact on cellular activities extends to the synthesis and packaging of ECV cargo, influencing variations in its composition throughout the day [7,8,9,10,11]. Disruptions to the circadian clock have been implicated in tumor development, progression, and responses to treatment across different types of cancers [12,13,14,15,16], which has a bearing on tailoring targeted therapeutic strategies to specific types of cancer [17,18]. Early detection and treatment of cancer is pivotal for enhancing patient outcomes. Therefore, sensitive and specific biomarkers are critical in early cancer detection and the monitoring of progression and treatment response. The sensitivity and specificity of the biomarkers currently available for cancer detection are inadequate, especially in accurately distinguishing between individuals at high or low risk for the disease [19,20,21,22,23]. Given their distinct composition and rich biological data, ECVs are an innovative and novel source of cancer biomarkers. One emerging barrier is their heterogeneity, with diverse subpopulations of tumor-derived ECVs playing varying roles in progression and metastasis [24,25,26,27,28].

Hepatocellular carcinoma (HCC) comprises over 90% of liver cancers and ranks as the fifth most common cancer and the third leading cause of cancer death worldwide, with 840,000 new cases and 780,000 deaths annually [29,30]. Underlying tissue alterations from hepatitis contribute to late-stage detection (85% at intermediate-to-advanced stages). Patients with early-stage disease are eligible for liver-directed therapies (including ablation), surgical resection, and liver transplantation [31]. Five-year survival with these potentially curative therapies can exceed 70%; in contrast, five-year survival after palliative therapies for advanced-stage disease is generally less than 5% [32,33]. Due to suboptimal surveillance methods for HCC (i.e., abdominal imaging with ultrasound, computed tomography, or magnetic resonance imaging) and persistent poor utilization and uptake among high-risk populations, most patients, unfortunately, present with advanced disease [32]. The only biomarker currently validated for clinical use for screening for HCC, alpha-fetoprotein (AFP), has marginal sensitivity and specificity (approximately 60 and 80%, respectively); not surprisingly, most societal guidelines recommend its use only in combination with abdominal ultrasound [32,34,35,36]. Other biomarkers that have been tested alone and in combination with AFP in large retrospective longitudinal studies, such as AFP-L3 and DCP, have similarly marginal performance characteristics [37]. As such, an urgent need exists for newer, more accurate, non-invasive biomarkers for early detection and diagnosis of HCC that do not necessarily need to be used in combination with abdominal imaging (which adds cost and impairs more widespread uptake, particularly across under-resourced settings). Several studies have indicated a close association between ECVs and the onset and progression of HCC. A thorough investigation into ECVs, including circadian influences, may reveal insights into tumor formation and metastasis mechanisms, offering novel approaches to early diagnosis and treatment [38,39,40].

## 2. Exosome Biogenesis

Traditional extracellular vesicles (ECVs) include exosomes (40–150 nm), microvesicles, and apoptotic bodies. Recent research has identified additional types, such as autophagic ECVs, stress-induced ECVs, and matrix vesicles [41,42,43]. The formation of exosomes occurs through a complex and highly regulated process, where intraluminal vesicles (IVLs) are created through inward budding endosomal membranes, with the sorting and enrichment of exosome components; exosomes are then disgorged from multivesicular bodies (MVBs) [44,45] (Figure 1).

Exosomes may release their material into the intracellular environment or internalize it into early endosomes. Early endosomes mature and form exosome-like vesicles, which can either be recycled and resecreted or degraded after fusion with a lysosome, supporting the recipient cell’s metabolism [45,46]. The genomic health of cells and distinct cell types can further influence the regulation of exosome biogenesis [47]. Exosomes are formed through two pathways: the endosomal sorting complex required for transport (ESCRT)-dependent pathway, and the ESCRT-independent pathway. The ESCRT-dependent pathway is considered the most important mechanism to sort cargoes and mediate membrane shaping for exosome formation. Numerous molecules, including the ESCRT complex, are involved in the sorting of ubiquitinated proteins into ILVs through this pathway. However, in the ESCRT-independent pathway, MBV and ILV formation occurs through the sphingomyelinase enzyme, using chaperones and tetraspanins (CD63 or CD81) to assist in cargo binding and membrane formation. Exosomal proteins, either luminal or surface-bound, enable subtyping with surface markers like CD9, CD63, and CD81, which are integral to exosome formation and function. Proteins such as GTPases, Annexins, and Rab GTPases facilitate endosomal processes and vesicle trafficking. Glycoproteins like β-galactosidase improve the exosome-targeting accuracy [48,49,50,51,52,53,54,55,56,57]. Exosomes also exhibit abundant nucleic acids, indicating an active sorting mechanism during their biogenesis [58], such as DNA, mRNA, and non-coding RNA (ncRNA) species [57,59]. MicroRNAs (miRNAs) stand out as one of the most abundant RNA species in exosomes, playing diverse roles in biological processes that are especially relevant to cancer, such as exocytosis, hematopoiesis, and angiogenesis, contributing to exosome-mediated cellular communication. Additional exosomal RNA species, like long ncRNA (lncRNA), circular RNAs (circRNAs), or small nuclear RNA, also impact biological processes, particularly influencing tumor development [60,61]. Exosomes affect target cells by binding to plasma membrane receptors or internalizing and releasing their contents, altering cellular processes and metabolism [45,46]. Current evidence suggests the selective packaging of ECV cargo, as shown by significant protein and RNA level variability compared to parental cells [62,63]. While the exact cargo-sorting mechanisms are unclear, RNA-binding proteins, Rab GTPases, and post-translational modifications, like ubiquitination and phosphorylation, are implicated [64]. Factors such as endoplasmic reticulum stress or phenotypic activation can influence ECV abundance and composition [65]. Current research focuses on the potential applications of these exosomal components as non-invasive biomarkers for disease diagnosis and prognosis.

## 3. The Role of Exosomes/ECVs in Oncogenesis

ECVs/exosomes play a critical role in the oncogenic process, facilitating the transfer of nucleic acids and proteins that disrupt cellular homeostasis and stimulate cancer initiation, progression, and metastasis. ECVs can carry oncogenic signatures that have the potential to transform recipient cells by altering gene expression and inducing malignant transformations [24,66,67].

Exosomes modulate immune responses, facilitating tumor cell evasion of immune surveillance and promoting metastatic spread. ECVs can transport immunosuppressive molecules that inhibit cytotoxic T cells and natural killer cells, allowing tumor cells to proliferate unchecked. Exosomes can also enhance metastasis by restructuring the extracellular matrix, which enables tumor cells to invade adjacent tissues and migrate to distant organs. Additionally, they play a vital role in the horizontal transfer of drug-resistance traits among cancer cell populations, spreading a chemoresistance that makes treatment more challenging. Clinically, exosomes can be extracted from body fluids like blood or urine and have the potential to monitor cancer non-invasively. The molecular composition of exosomes reflects the pathological state of their cells of origin, providing insights that are crucial in early detection, tracking disease progression, or adjusting therapeutic strategies. Through the analysis of exosomal biomarkers, clinicians can identify specific molecular signatures indicative of cancer presence, evaluate the aggressiveness of the disease, identify drug resistance, and monitor therapeutic response. This capability is essential in implementing personalized treatment plans that optimize therapy effectiveness and improve overall patient prognosis, making exosomes a cornerstone of modern oncological research and treatment strategies [24,48,66,67,68,69]. Table 1 aggregates the changes in exosomal ncRNA levels across various cancer types, revealing how these molecules could serve as biomarkers for diagnosis, prognosis, and treatment monitoring, particularly highlighting the extensive range of applications in diverse cancer contexts.

### 3.1. Hepatic-Cell-Derived ECVs: Cargo and Functions

Approximately 80% of the liver’s volume is made up of hepatocytes, which are crucial in physiological processes, while 6.5% consists of non-parenchymal cells, like liver sinusoidal endothelial cells (LSECs), hepatic stellate cells (HSCs), cholangiocytes, and Kupffer cells. Minority cell populations support hepatocytes and maintain the hepatic environment. The liver’s lobular structure promotes effective bidirectional intercellular communication and molecular information transfer through ECVs (Figure 2). For example, hepatocyte-derived ECVs carry arginase-1, which modulates endothelial cell function, and sphingosine-1-phosphate, which promotes liver regeneration. LSEC-derived ECVs modulate hepatic stellate cell activity, and HSC-derived ECVs may facilitate HCC progression. Cholangiocyte-derived ECVs affect bile acid homeostasis and promote healing. This encapsulates the varied currently known roles and cargo of hepatic-cell-derived ECVs in liver function and disease [93,94,95,96,97,98,99,100,101].

In addition to playing a direct role in HCC, ECVs from hepatocytes and adipocytes carry specific cargoes that enhance steatosis and immune activation, driving the progression of chronic liver diseases like non-alcoholic fatty liver disease and alcoholic liver disease and ultimately increasing the downstream risk for HCC [102]. For example, hepatocyte-derived ECVs contain miRNAs such as miRNA-122, miRNA-192, and miRNA-128-3p, as well as mitochondrial DNA, influencing inflammation and metabolic dysregulation in non-alcoholic liver disease [103,104]. In alcoholic liver disease, hepatocyte-derived ECVs carry CD40 ligand and mitochondrial DNA, promoting inflammation and immune cell activation [105]. However, adipocyte-derived ECVs carry miRNA-99b target hepatocytes to promote steatosis [106]. Additionally, circulating ECVs in HCC display altered expression of surface markers, including Annexin V, EpCAM, ASGR1, and CD133, offering a panel of markers to distinguish HCC from cholangiocarcinoma and other malignancies [107,108].

### 3.2. Exosomes in HCC Initiation, Progression, Metastasis, and Angiogenesis

In HCC, ECVs, particularly exosomes, play a fundamental role in initiation, progression, and metastasis. These vesicles facilitate complex signal transductions within the tumor microenvironment, significantly enhancing cellular communication between tumor cells and surrounding non-tumor cells. ECVs transport a variety of bioactive molecules, including miRNAs and proteins, which are instrumental in promoting cellular transformations, inflammatory responses, and adaptive mechanisms essential to tumor survival and expansion. Through their cargo, ECVs upregulate inflammatory cytokines and support adaptive responses under hypoxic conditions, contributing to tumor growth and the spread of cancer cells. They also induce critical processes such as epithelial–mesenchymal transition (EMT) and manipulate the immune landscape, enhancing the tumor’s invasiveness and capacity to evade immune detection [38,109]. Additionally, the acidic microenvironment in HCC further stimulates ECVs to strengthen the expression of miRNAs that promote cell proliferation and invasion, intensifying the aggressiveness of cancer [110]. Table 2 contains selected examples and emphasizes the various roles of ECVs in the context of HCC.

### 3.3. ECVs and Tumor Microenvironment

ECVs, including exosomes, are critical mediators within the tumor microenvironment, which is essential for tumor progression and metastasis. In HCC, the tumor microenvironment often becomes acidic and hypoxic, influencing the behavior of ECVs. This acidic environment arises due to inadequate perfusion in early tumors, leading to hypoxia and a reliance on anaerobic metabolism, which elevates lactate production and acidifies the tumor microenvironment [131]. This metabolic reprogramming, known as the “Warburg effect”, occurs even in the presence of sufficient oxygen. Due to these acidic conditions, ECV levels increase and enhance the proliferation, migration, and invasion of recipient cells [132]. Moreover, hypoxia modulates the expression of cell surface receptors and ceramides through hypoxia-inducible factors, further facilitating tumor metastasis [133,134].

ECVs and various non-immune cells in the tumor microenvironment, such as cancer-associated fibroblasts (CAFs), endothelial cells, and adipocytes, play a crucial role in tumor progression and metastasis. CAFs, arising from diverse cell origins, including fibroblasts and mesenchymal stem cells, are pivotal in the microenvironment, as they deposit extracellular matrix proteins and facilitate tumor invasion. Studies have shown that ECVs released by highly metastatic HCC cells contain miRNA-1247-3p, which can transform normal fibroblasts into CAFs, enhancing metastasis through the secretion of pro-inflammatory cytokines. Conversely, CAF-derived ECVs with reduced levels of miRNA-150-3p have been found to suppress the migration and invasion of HCC cells [135,136]. Cargo from endothelial-cell-derived ECVs facilitate angiogenesis, enhancing HCC proliferation and metastasis [86,120,122,137,138]. Additionally, adipocytes within the tumor microenvironment, particularly cancer-associated adipocytes, contribute to tumor progression by creating a hypoxic environment, promoting inflammation, and remodeling the extracellular matrix. These interactions between ECVs and the tumor microenvironment highlight the complexity of HCC pathogenesis, emphasizing the need for targeted therapeutic strategies. Moreover, ECVs’ role within the tumor microenvironment is crucial in advancing both cancer treatment and diagnostics.

### 3.4. Diagnostic Biomarkers in HCC

Due to the absence of reliable biomarkers, the early diagnosis of HCC is challenging, with most cases identified at advanced stages [34]. There is growing interest in potentially using “liquid biopsies” for the early detection of HCC [139,140]. Although several hypermethylated target genes identified in circulating cell-free DNA have been shown to be specifically associated with HCC, their expression in the peripheral blood is very low, particularly in the setting of early-stage disease [139]. Non-coding microRNAs and long non-coding RNAs, which play regulatory roles in various diseases and cancers, have also been investigated as potential biomarkers for HCC. Although their utility is somewhat limited due to differences across ethnic populations and underlying etiologies of HCC, studies have suggested that when used in combination with each other, as well as with AFP, they can show an excellent diagnostic performance (i.e., area under the curve (AUC) > 90%) particularly when enriched within exosomes (rather than when detected in exosome-depleted serum fractions) [141,142,143,144].

ECVs are promising diagnostic markers for cancer due to their lipid bilayer, which preserves crucial biomolecules and ensures stability in bodily fluids, enabling minimally invasive and more reliable detection [145]. This resilience ensures that both fresh and long-term-stored exosome samples maintain important tumor information for analysis [67,146,147,148].

Exosomes in HCC have potential clinical applications, as they promote immune escape, angiogenesis, metastasis, and tumor invasion. They can also confer drug resistance and modulate immunotherapy responses. Exosomes may serve as biomarkers for detecting and monitoring HCC, highlighting their relevance to targeted therapies and personalized medicine (Figure 3). In addition to total circulating ECV markers, specific cell membrane proteins can integrate into secreted ECV membranes [149]. These cell-type-specific surface proteins aid in the immunoaffinity-based isolation of ECVs from particular cells or tissues, enhancing disease biomarker detection sensitivity and specificity. Goetzl et al. have pioneered a novel method in the field of non-invasive biomarkers by isolating fetal central-nervous-system-derived exosomes from maternal blood. This innovative technique enables the detection and characterization of fetal brain injuries through a non-invasive approach, significantly advancing prenatal diagnostics and monitoring [150,151,152,153]. Similarly, research has demonstrated the efficacy of using asialoglycoprotein receptor 1 (ASGR1) to purify hepatocyte-derived exosomes, significantly improving liver disease biomarker detection [154]. In the context of HCC, isolating HCC-derived ECVs is critical in developing precise biomarkers, offering a targeted approach compared to total circulating ECVs. Mass spectrometry of exosomal proteins from HCC cell lines has identified approximately 1400 proteins that facilitate intercellular communication and correlate with clinical parameters, such as tumor size, TNM stage, portal vein tumor thrombosis, and overall survival. This targeted isolation provides superior diagnostic and prognostic capabilities compared to conventional biomarkers like alpha-fetoprotein (AFP), enhancing the specificity and sensitivity of disease detection and enabling more effective personalized treatment strategies for HCC patients [155,156]. Also, it has been shown that the quantity of serum ECVs notably increases during the cirrhotic stage and early stages of HCC compared to healthy liver tissue [157,158], suggesting a potential role of HCC-derived ECVs in the early detection of HCC [148].

Lipids play a dual role in exosomes. They act as structural components of the exosomal membrane and protect the cargo from peripheral degradation. They are essential contributors to exosome formation and release. It is common to see enrichment of specific lipids in ECVs, creating a “lipid signature” that could be used for diagnosis/prognosis in HCC [159,160]. For example, HepG2/C3a-cell-derived ECVs have specific lipid profiles with higher free cholesterol, ceramides, phosphatidylserine, and sphingomyelin but lower phosphoinositide levels [161,162,163]. A clinical study showed significant differences in lipid profiles between HCC and non-HCC patients, identifying higher levels of sphingosines, dilysocardiolipins, and (O-acyl)-1-hydroxy fatty acids with HCC, highlighting their potential as non-invasive diagnostic biomarkers [164].

Several studies have highlighted the potential of exosomal miRNAs as early diagnostic biomarkers from HCC cells. Exosomal miRNAs offer superior stability and reduced susceptibility to interference from other blood components, enhancing their biomarker potential compared to serum-free miRNAs [165,166,167]. Hyo et al. identified exosomal miRNA-10b-5p as a promising biomarker (AUC, 0.934; sensitivity, 90.7%; specificity, 75.0%; cutoff, 1.8-fold). Another study advocated for the combination of exosomal miRNA-466-5p and miRNA-4746-5p, achieving an AUC of 0.947 (CI, 0.889–0.980; sensitivity, 81.8%; specificity, 91.7%) [167,168,169]. Decreased miRNA-638 levels in HCC patient serum correlate with adverse tumor characteristics, while elevated levels correlate with improved survival [170]. Additionally, exosomal miRNA-320d levels in serum samples could be used to distinguish HCC patients from healthy controls. Decreased exosomal miRNA-320d levels were associated with advanced tumor stage and positive lymph node metastasis and, therefore, with shorter overall and disease-free survival, indicating poor prognosis in HCC [171].

While the studies on exosomal DNA in HCC are limited, researchers are currently focusing on its potential in tumor diagnosis [38,172]. Yan et al. discovered a significant elevation in the cell-free DNA levels of HCC patients compared to non-HCC patients. The isolation and analysis of disease-specific ECVs represent significant advancements in biomarker research, improving the accuracy and efficacy of non-invasive diagnostics and personalized medicine.

### 3.5. ECVs in Immunotherapy and Therapy Resistance

Exosomes and ECVs significantly influence HCC therapy through their dual roles in immune modulation and drug resistance. These vesicles regulate immune responses that are crucial to personalized cancer treatment, notably via interactions with the PD-1/PD-L1 axis [173,174]. Additionally, ECVs can act as precise drug delivery vehicles, enhancing the targeting and dosage customization of treatments like Doxorubicin and Paclitaxel, thus reducing toxicity while improving effectiveness [175,176,177,178,179].

Beyond PD-1/PD-L1 inhibition therapy, ECVs also impact the efficacy of other immune checkpoint inhibitors. Tumor-derived ECVs can carry ligands for these checkpoints, modulating the immune environment and contributing to immune evasion by tumors [180,181,182].

Although primarily used for hematologic malignancies, recently, CAR T-cell therapy has been explored for solid tumors, like HCC. ECVs can modulate the tumor microenvironment, enhancing CAR T-cell persistence and functionality [183]. For instance, mesenchymal-stem-cell-derived ECVs have been shown to enhance CAR T-cell anti-tumor activity by modulating the immune environment. Additionally, ECVs from HCC cells can carry immunosuppressive molecules, such as TGF-β and IL-10, inhibiting CAR T-cell activity [182,184,185]. Engineering ECVs to carry pro-inflammatory cytokines or immune checkpoint inhibitors can potentially counteract this suppression and enhance the efficacy of CAR T-cell therapy.

Dendritic-cell-derived ECVs are particularly promising as immunotherapeutic agents. These ECVs contain immunostimulatory components, functioning as antigen-presenting entities. They can promote an immune-cell-dependent tumor rejection response by enhancing the activation of CD8+ T cells and remodeling the tumor microenvironment. Tumor-derived ECVs also stimulate anti-tumor immune responses and deliver tumor-associated antigens to dendritic cells, enabling efficient antigen presentation on major histocompatibility complex (MHC) molecules to T cells. Inspired by CAR T-cell therapy, researchers have used dendritic-cell-derived ECVs to present MHC–antigen complexes, triggering effective anti-tumor immunity. These engineered ECVs can offer an effective alternative to CAR T cells by promoting T-cell binding to cancer cells [183,186].

Moreover, engineered ECVs are emerging as a promising therapeutic strategy for tumor immunomodulation. For example, exoASO-STAT6 uses ECVs to deliver antisense oligonucleotides that disrupt STAT6 signaling in tumor-associated macrophages. This method has shown strong anti-tumor activity in preclinical models and is currently being tested in clinical trials for advanced HCC [187,188,189].

Another critical function of ECVs is mediating resistance to therapies, particularly chemotherapy. They transport molecules, such as specific proteins or RNA, that induce drug resistance, spreading it throughout the tumor. Identifying biomarkers within exosomes that signal resistance allows clinicians to develop tailored treatment plans. These plans might include agents that sensitize tumors to intended therapies, boosting the efficacy of standard treatments without increasing side effects [84,190,191,192,193,194,195].

Exosomes’ capability to carry both water-soluble and lipid-soluble drugs highlights their potential to expand the effectiveness of various cancer therapies. Their versatility and compatibility with biological systems make them valuable in precision medicine, aiming to customize cancer treatments to individual patient profiles for better outcomes [196,197].

Moreover, ECVs have the capacity to enhance and modulate immune responses, positioning them as a promising strategy for novel vaccine formulations. ECVs can potentially activate granulocytes or natural killer cells and interact with CD8, CD4, and B cells, eliciting antigen-specific immune responses. The combination of ECVs with antigens may induce a humoral immune response comparable to that elicited by the antigen alone. This finding suggests that ECVs could serve as vaccine adjuvants, enhancing the efficacy of vaccines [187,188,198].

Recent clinical trials by Escudier et al. and Morse et al. have focused on dendritic-cell-derived exosomes, demonstrating their potential in advancing cancer therapies. These studies show promising results in terms of both safety and efficacy. The positive outcomes indicate the feasibility of integrating exosome-based approaches into standard cancer treatment protocols, leading to more personalized and effective strategies [196,197,199]. Table 3 provides additional details on the role of ECVs in enhancing immunotherapy and managing therapy resistance, showcasing their multifaceted contributions to improving HCC treatment. Exosomes and ECVs are crucial in advancing HCC therapy. Their functions in immune modulation, drug delivery, and overcoming resistance position them at the forefront of personalized cancer treatment strategies, with ongoing research and clinical trials continually revealing their full potential.

## 4. The Molecular Basis of Circadian Rhythms in Mammals

Mammals synchronize their circadian rhythms primarily with the light–dark cycles in their environment, a process mediated by ocular photoreception that relays signals to the suprachiasmatic nucleus of the hypothalamus. This region coordinates the synchronization of circadian clocks across the body’s various tissues. Peripheral clocks are entrained by signals such as humoral cues, metabolic factors, and body temperature fluctuations [201,202,203]. At the molecular level, the circadian clock is driven by autoregulatory transcription–translation feedback loops (TTFLs) involving transcription factors like brain and muscle ARNT-like 1 (*BMAL1*) and circadian locomotor output cycles kaput (*CLOCK*). These proteins form the CLOCK-BMAL1 heterodimer that binds to E-Box sequences, promoting the expression of clock-regulated genes, including Period (PER) and Cryptochrome Circadian Regulator (CRY). PER and CRY proteins inhibit *BMAL1* activity, and their stability is regulated by ubiquitin ligases and kinases, establishing a new oscillatory cycle. Additional regulators such as REV-ERBs and retinoic acid receptor-related orphan receptors create another feedback loop by repressing or activating *BMAL1* expression, respectively [204,205] (Figure 4). Other molecular oscillators can also function independently of the transcription-based clock in various species [201,206].

Circadian disruptions stem from environmental, genetic, and pathobiological factors. They influence the body’s internal clocks, disturbing critical physiological functions, such as sleep, alertness, motor skills, body temperature regulation, urinary system functionality, hormone secretion, immune responses, cytokine release, and cell cycle progression [207,208,209]. Shift work, jet lag, exposure to artificial light at night, irregular meal timings, alcohol consumption, and late-night physical activities are the main environmental factors contributing to circadian misalignment [209]. Genetic mutations in circadian clock genes and neurodevelopmental genetic differences further disrupt circadian function, increasing the risk of various diseases, including cancer [12,14,205,210,211]. Aging, obesity, hyperglycemia, and inflammatory states can exacerbate the dysregulation of circadian rhythmicity, augmenting cancer risk [212,213,214].

### 4.1. Circadian Regulation of ECVs

Multiple lines of evidence support the circadian clock’s influence on intercellular communication via ECVs, particularly exosomes. Rhythmic gene expression and protein abundance indirectly affect the temporal-specific loading of proteins in small ECVs, while the circadian clock directly governs the abundance of targeted cargo proteins in exosomes through the controlled expression of sorting proteins in the endosomal pathway. Studies have shown that ECV quantity exhibits time-dependent changes, indicating circadian variation [215], while the circadian clock directly influences the cargo content of exosomes [200]. Additionally, disruptions in circadian rhythms, such as those induced by night shift work, can alter the exosomal cargo, impacting metabolic health [216]. Plasma ECVs display dynamic changes in size distribution throughout the day, with implications for intercellular communication [217]. Overall, the circadian regulation of ECV characteristics, including size, concentration, and cargo composition, underscores the critical influence of the circadian clock in ECV biology for diagnostics and therapeutics. These rhythmic changes, governed by the circadian clock, have broad implications for intercellular communication and physiological processes, including those relevant to cancer. Understanding the role of the circadian clock in shaping ECV behavior in cancer contexts is crucial for developing time-sensitive strategies to diagnose, monitor, and treat various malignancies, including HCC, by optimizing treatment efficacy and patient outcomes. The impact of circadian rhythms on exosome secretion also suggests that aligning therapeutic interventions with these natural cycles could enhance treatment effectiveness and reduce adverse effects [8,200].

Table 4 summarizes studies on circadian influence on ECVs, showing rhythmic variations in size, concentration, and cargo composition.

### 4.2. Circadian Clock and HCC

Liver metabolism is intricately governed by circadian rhythms, adjusting to changes in feeding times and dietary patterns [222,223,224,225,226]. Central to this regulation are crucial circadian genes such as CRY, BMAL1, and CLOCK, which play pivotal roles in managing glucose levels and fat metabolism [227,228]. Disruptions in the liver’s internal clock, whether from genetic factors or external influences like jet lag, significantly affect metabolic processes, including bile acid regulation. Studies link circadian rhythm disruptions to increased cancer risk, with abnormal gene expression being common in tumors [84,217,229]. Night shifts and rotating work schedules are associated with higher cancer incidence and were recognized as a probable carcinogen by the International Agency for Research on Cancer in 2007, which was reaffirmed in 2019 [230]. Circadian clocks regulate key cellular functions like growth, apoptosis, and DNA repair, impacting tumor behavior [14,205,231,232,233,234,235]. The disruption of circadian rhythms due to chronic jet lag has been linked to accelerated liver carcinogenesis, as well as increased susceptibility to non-alcoholic fatty liver disease, leading to the spontaneous development of HCC [236]. Recent findings have linked variations in circadian clock genes with survival rates and clinical outcomes in HCC patients [237]. The development of small-molecule modulators targeting the core circadian clock offers a new promising approach in cancer therapy, potentially leading to innovative treatments [238,239,240]. Research has consistently underscored the role of clock genes in the molecular mechanisms of HCC [241], driving focused efforts to develop therapeutic strategies that target these proteins to effectively treat HCC [241,242,243,244].

### 4.3. The Impact of the Circadian Clock on Cancer Progression via ECVs

Melatonin, a key signal of darkness, plays a vital role in regulating sleep–wake cycles and shows promise in treating conditions such as sleep disorders and jet lag, as well as enhancing cancer therapies by improving the effects of chemotherapy [245,246]. Melatonin increases the properties of exosomes, potentially enhancing their therapeutic efficacy, such as in reducing inflammatory factors and immune evasion markers like PD-L1 in cancer therapy.

Cheng et al. observed a suppression of PD-L1 expression in macrophages co-cultured with exosomes from HCC cells treated with 0.1mM melatonin, while they observed an increase in PD-L1 expression and cytokine levels in macrophages co-cultured with untreated HCC cells. This led to the conclusion that melatonin-treated exosomes effectively reduced PD-L1 expression and cytokine secretion. Therefore, manipulating circadian rhythms through exosomes could offer innovative strategies for combating inflammation and immune evasion in HCC, inhibiting STAT3 activation and potentially improving therapeutic outcomes for patients with this challenging disease [247].

In addition to the emerging evidence in HCC, there are additional data pointing to a more overarching role of circadian rhythm in cancers in general. The circadian clock regulator modulates tumorigenesis [248]. To be precise, BMAL1, an essential transcription factor regulating numerous clock target genes [249,250], promotes metastasis in colorectal cancer by enhancing exosome secretion, illustrating its crucial role in cancer progression tied to circadian regulation [17]. Research using a mouse model of nocturnal shift work showed changes in the intestinal flora and plasma ECV components affecting clock genes [251]. SIRT1, interacting with CLOCK-BMAL1, regulates PER2 and exosome secretion, impacting the tumor microenvironment and progression in breast and ovarian cancer [252,253,254,255,256,257,258], and is crucial in tumor dynamics and metastasis [259]. ECVs regulate key signaling pathways like GSK-3 and AMPK via non-coding RNAs, affecting the circadian regulators CRY and BMAL1/CLOCK [258,260,261,262]. Exosomes also modulate circadian gene expression through post-transcriptional mechanisms, playing a role in rhythmicity and offering therapeutic potential [9,263]. Additionally, exosomal miRNAs like miRNA-7239-3p from microglia influence circadian genes, impacting conditions such as glioma progression [264]. An interesting proteomic study on medulloblastoma-derived exosomes detected that the presence of proteins influenced by the circadian clock, such as transcription factors like HNF4A, can impact pathways associated with various malignancies. HNF4A, a circadian-regulated transcription factor, modulates metabolic processes and adjusts tissue-specific circadian networks by transrepressing CLOCK-BMAL1 [244,265,266]. In HCC, the fetal variant of P2-HNF4α, typically upregulated, significantly alters the expression of BMAL1, influencing cancer progression dynamics. The presence of these proteins in exosomes suggests that circadian-influenced exosomal proteins could serve as non-invasive biomarkers for early cancer detection. Table 5 summarizes a structured overview of selected circadian biology that intersects with cancer progression through the lens of ECV dynamics.

## 5. Conclusions

This review highlights the significant interplay between ECVs, especially exosomes, and the development and progression of HCC, as well as the potential clinical roles of ECVs in detection and treatment. Disruptions in circadian rhythms may alter ECV function, potentially initiating or exacerbating HCC. These vesicles facilitate crucial intercellular communication and may modify circadian rhythms in recipient cells, influencing disease progression.

Exosomal cargo-sorting mechanisms in HCC involve RNA-binding proteins, Rab GTPases, and post-translational modifications. For example, the overexpression of VpsA4, an ATPase essential to the ESCRT pathway, modifies exosomal miRNA levels in HCC cells [276]. Rab GTPases and RNA-binding proteins are also critical in exosome biogenesis and influence cell signaling and tumor progression [277]. Moreover, exosomal lipids are also essential in exosome biogenesis and function [278]. Further research into the sorting mechanisms of exosomal cargo is crucial. Understanding the roles of RNA-binding proteins, Rab GTPases, and post-translational modifications will enhance our comprehension of how exosomes influence disease processes. Additionally, the effect of circadian clock genes on the tumor microenvironment, particularly regarding immune cell infiltration and overall cancer progression, underscores the importance of exploring how circadian genes impact exosome-mediated communication.

ECVs facilitate intercellular communication through their diverse cargo. Quantitative proteomic analyses across various cell lines, including hepatocytes, have identified unique ECV protein profiles that can predict small changes in serum [279]. In addition, exosomal biomarkers, such as miRNAs and circRNAs, showed higher sensitivity and specificity than AFP, a traditional biomarker widely used in clinical practice for detecting HCC [34,51]. To date, several methods for exosome isolation have been developed to enhance precision and throughput, including microfluidics-based isolation, density gradient ultracentrifugation, immunoaffinity capture, size-exclusion chromatography, polymer-based precipitation, ultrafiltration, field-flow fractionation, and differential ultracentrifugation. Immunoaffinity capture techniques use antibodies targeting specific exosome markers, enabling selective isolation from specific cell types like hepatocytes [280,281].

Exosomal biomarkers are stable in bodily fluids and offer non-invasive detection methods, providing detailed insights into tumor dynamics, including growth, metastasis, and treatment responses [24]. However, despite these advantages, the clinical implementation of exosomal biomarkers faces several challenges. The heterogeneity of exosomes, derived from various cell types, leads to variability in their content and function, complicating the isolation of tumor-specific exosomes [56,282]. Current isolation methods, such as ultracentrifugation and immunoaffinity capture, vary in efficiency and specificity, underscoring the need for standardized protocols to ensure reproducibility and reliability in clinical settings [281]. Accurate quantification and characterization of exosomal biomarkers are challenging due to their small size and complex cargo, necessitating the development of more precise and sensitive analytical technologies [52]. Furthermore, extensive clinical trials and validation studies are required to establish the efficacy of exosomal biomarkers in routine clinical practice. Regulatory approval for new diagnostic tools based on exosomal biomarkers involves rigorous evaluation and adherence to standards [52,283]. Addressing these challenges can highlight the potential advantages of exosomal biomarkers and support their successful clinical implementation in HCC diagnosis and treatment [284].

Innovative techniques for non-invasive biomarkers, such as isolating cell-specific exosomes, could offer promising avenues for early disease detection and intervention. One study showed that employing ASGR1 to purify exosomes derived from hepatocytes improved the detection of liver disease biomarkers [154]. To further refine this technique, focusing on additional liver-specific proteins such as hepatocyte-specific antigens, liver-specific transport proteins, and liver fatty-acid-binding proteins could significantly enhance the precision and efficacy of these diagnostic tools.

Furthermore, aligning medical treatments with the patient’s body clock through chronotherapy or pharmacological manipulations of the circadian clock could retard tumor growth [205,285]. Ideally, insights from the circadian regulation of exosome release and composition should be considered to optimize treatment efficacy. Therefore, the time of blood sample collection should be considered covariable in the design of clinical studies. This parameter may promote insights into how the circadian clock influences the biology of cancer via exosomal proteins, which are crucial mediators in intercellular communication.

Understanding the complex interplay among exosomal components, cancer biology, and the circadian rhythm is essential. This relationship forms the basis of a rich field of study that promises to broaden our understanding of the mechanisms underlying malignancy and innovate diagnostic and therapeutic strategies. The ultimate goal is to leverage this knowledge to develop more personalized, timely, and effective treatment approaches, improving clinical outcomes for patients with HCC.

Future research should integrate these findings into clinical practice to enhance the precision and efficacy of HCC treatments. Leveraging ECVs for early disease detection, targeted drug delivery, and immune modulation in HCC provides significant advantages, including higher sensitivity and specificity in diagnostics, improved therapeutic outcomes, and reduced systemic toxicity compared to existing methods. However, addressing challenges such as exosome heterogeneity, isolation, and standardization is critical. Developing standardized protocols for exosome isolation and characterization, elucidating the mechanisms of ECV biogenesis and cargo sorting, and conducting comprehensive clinical trials are essential steps to validate the clinical utility of exosome-based therapies.

## Figures and Tables

**Figure 1 cancers-16-02552-f001:**
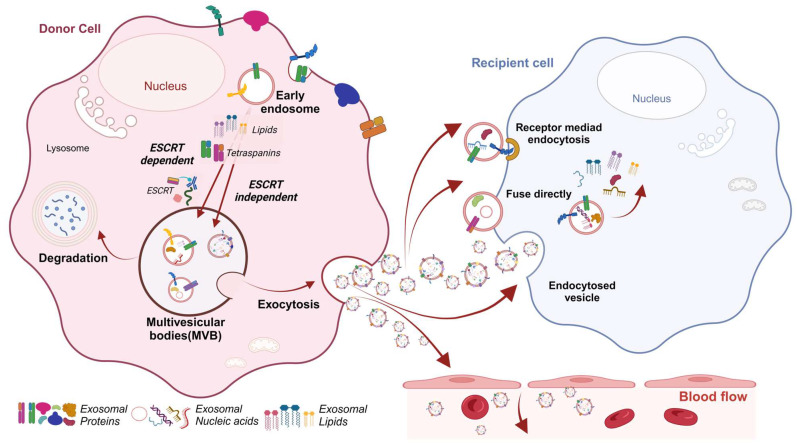
Exosome biogenesis. The primary pathway of exosome biogenesis involves the creation of multivesicular bodies (MVBs) within endosomes, leading to exosome secretion. During endocytosis, early endosomes transform into MVBs, generating intraluminal vesicles (ILVs) through the inward budding of their membranes. These MVBs may either merge with lysosomes for degradation or move to the cell surface to release exosomes. This release process is facilitated by RAB GTPases and SNARE complexes. Once exosomes are released, extracellular vesicles (ECVs) can interact with target cells through ligand–receptor binding, endocytosis, or membrane fusion, allowing the delivery of their cargo into the cytoplasm of the recipient cell. The molecular content of ECVs plays a crucial role in regulating various functions in target cells, including intracellular signaling, gene regulation, and metabolism, and can also be directly released into biological fluids. Created with BioRender.com.

**Figure 2 cancers-16-02552-f002:**
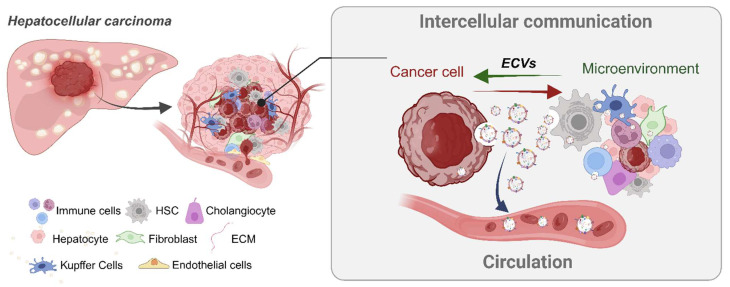
Impact of extracellular vesicles (ECVs) in the hepatocellular carcinoma (HCC) tumor microenvironment. ECVs are released by both tumor cells and other surrounding cells into the tumor microenvironment and the circulatory system. These ECVs transport a variety of bioactive molecules that contribute to tumor growth, metastasis, and intercellular communication. By mediating these exchanges, ECVs play a key role in shaping the tumor microenvironment, potentially influencing the progression of cancer and the behavior of distant cells through the molecules that they carry. ECVs, extracellular vesicles; HSCs, hepatic stellate cells; ECM, extracellular matrix. Created with BioRender.com.

**Figure 3 cancers-16-02552-f003:**
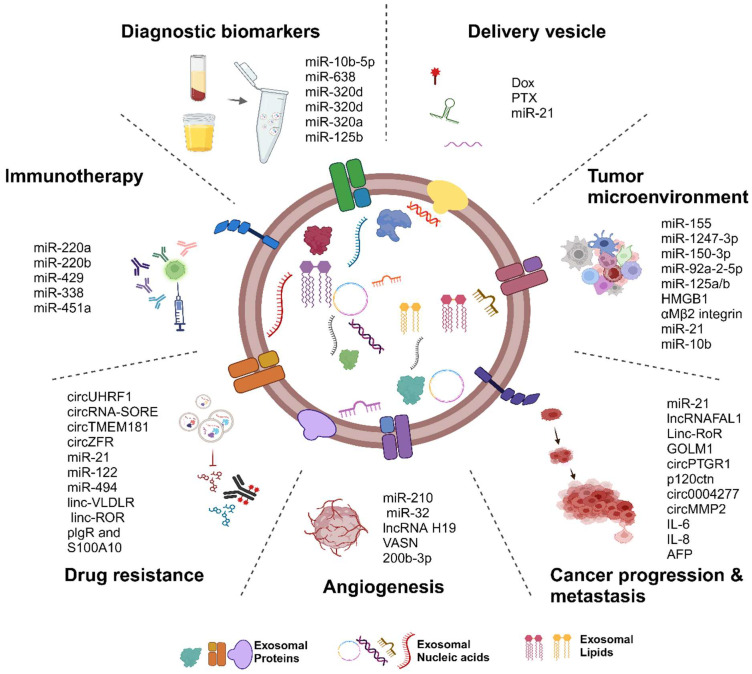
Roles of exosomes in HCC. This diagram demonstrates the multiple functions of exosomes in HCC, highlighting their roles in facilitating immune escape, promoting angiogenesis, driving metastasis, and supporting tumor invasion. Exosomes from other sources, like stem cells and macrophages, can also influence HCC by conferring drug resistance and modulating responses to immunotherapy. These exosomes have additional potential as biomarkers for detecting and monitoring HCC, shedding light on disease progression and offering new pathways for treatment. The ability of exosomes to mediate intercellular communication within the tumor microenvironment emphasizes their relevance to targeted therapies and personalized medical approaches. Created with BioRender.com.

**Figure 4 cancers-16-02552-f004:**
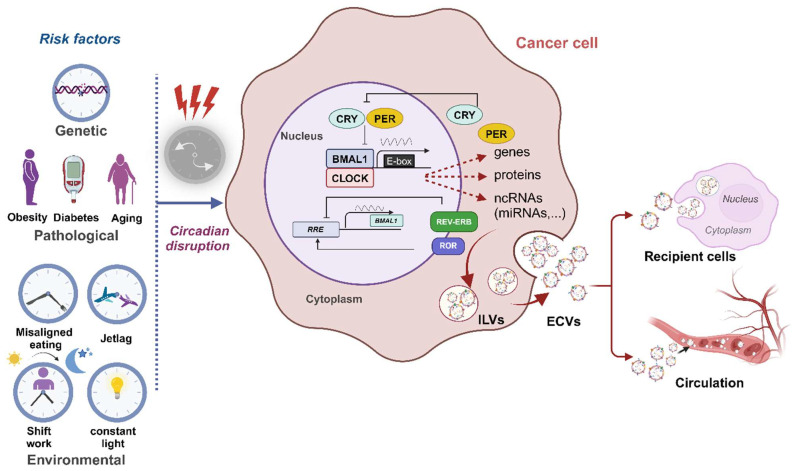
Circadian clock’s impact on ECV characteristics in cancer. In the molecular clock, CLOCK and BMAL1 rhythmically bind to E-boxes, activating clock-controlled genes and other clock components. Their activity is regulated by PER and CRY proteins, which inhibit CLOCK-BMAL1 upon entering the nucleus. A second feedback loop involves *BMAL1* transcription, which is repressed by REV-ERB and activated by ROR. The circadian clock plays a crucial role in shaping the properties of ECVs, which are essential for cellular communication and metabolic regulation. Disruptions in circadian rhythms, resulting from factors such as genetic mutations in clock genes; environmental stresses like irregular eating times, jet lag, shift work, and constant light exposure; and pathological conditions like aging, diabetes, and obesity can drive cancer development and progression. These disturbances in circadian patterns significantly affect ECV characteristics, including size, concentration, and cargo composition, emphasizing the critical relationship among circadian rhythms, ECV biology, and cancer dynamics, whether through circulation or interactions with recipient cells. Created with BioRender.com.

**Table 1 cancers-16-02552-t001:** Exosomal ncRNAs as potential cancer biomarkers in diagnosis and progression.

Exosomal ncRNA	Type of Disease	Source of Exosome	Expression	References
miRNA-21, miRNA-1224, miRNA-1229, miRNA-1246, miRNA-150, miRNA-21, miRNA-223, miRNA-23a	Colorectal cancer	Blood	↑, ↑, ↑, ↑, ↑, ↑, ↑, ↑	[70,71]
miRNA-21, miRNA-105, miRNA-372	Breast cancer	Blood	↑, ↑, ↑	[72,73]
miRNA-373	Triple-negative breast cancer	Blood	↑	[74]
miRNA-21, miRNA-141, miRNA-200a, miRNA-200c, miRNA-200b, miRNA-203, miRNA-205, miRNA-214	Ovarian cancer	Blood	↑, ↑, ↑, ↑, ↑, ↑, ↑, ↑	
miRNA-17-5p, miRNA-21, miRNA-106a, miRNA-106b	Pancreatic cancer	Blood	↑, ↑, ↑, ↑	[75,76]
miRNA-184	Tongue cancer	Blood	↑	[77]
miRNA-21, miRNA-500, mi-RNA-26a, mi-RNA-26c, miRNA-224, miRNA-665, miRNA-10b-5p, miRNA-18a-5p, miRNA-215-5p, miRNA-940mi-RNA-199a, miRNA-18a, miRNA-221, miRNA-222	Hepatocellular carcinoma	Blood	↑, ↓, ↓, ↓, ↑, ↑, ↑, ↑, ↑, ↑, ↑, ↑, ↑	[78,79,80,81,82]
miRNA-125a, miRNA-200a	Oral squamous cell carcinoma	Blood	↓	[83]
circ-DB, circCUHRF1, circTMEM4A, circ-100338, circ-0051443	Hepatocelluar carcinoma	Blood	↑, ↑, ↑, ↑, ↓, ↓	[84,85,86,87]
circ_0047921, circ_0056285, circ_0007761	Non-small cell lung cancer	Blood	↓, ↓, ↑	[88]
circ_0001439, circ_0001492, circ_0000896	Lung adenocarcinoma	Blood	↑, ↑, ↑	[89]
circMYC	Nasopharyngeal carcinoma	Blood	↑	[90]
circ_0047921, circ_0056285, circ_0007761	Non-small cell lung cancer	Blood	↓, ↓, ↑	[88]
hsa_circ_0055202, hsa_circ_0074920 hsa_circ_0043722	Glioblastoma	Blood	↑, ↑, ↑	[91]
circPDLIM5	Prostate cancer	Urine	↑	[92]

↑ up ↓ down.

**Table 2 cancers-16-02552-t002:** Role of ECVs in HCC carcinogenesis.

Exosomal Component	Effect	Mechanism of Impact	References
miRNA-93	Tumor proliferation	Enhances the growth of HCC cells	[111,112]
PDGFRα, Hedgehog ligands	Fibrogenesis, angiogenesis	Promotes tissue remodeling and blood vessel formation	[64]
miRNA-1247-3p	Lung metastasis	Converts cells to cancer-associated fibroblasts and increases pro-inflammatory cytokines (IL-6, IL-8)	[18]
Circ-0004277, miRNA-136-5p	EMT, tissue invasion	Induces epithelial-to-mesenchymal transition in surrounding cells	[113,114]
miRNA-21, miRNA-10b	Tumor proliferation, metastasis	Regulation of the tumor suppressor gene PTEN through modulation of Tet methylcytosine dioxygenase expression. Increases HIF-1α mRNA and promotes survival under hypoxia	[110]
miRNA-155	Inflammation	Intensifies IL-6 and IL-8 levels and enhances inflammatory response	[115,116]
GOLM1	Tumor occurrence, metastasis	Activates GSK-3β/MMP signaling, a potential biomarker	[38,117]
miRNA-320a, miRNA-451a	Anti-tumorigenic effects	Inhibits proliferation, migration, angiogenesis	[118,119]
LncRNA H19	Enhanced angiogenesis	Increases VEGF secretion and VEGF-R1 production	[120]
miRNA-32	Angiogenesis	Suppresses PTEN and activates PI3K/Akt pathway	[110]
VASN	Tumor development, angiogenesis	Stimulates endothelial cell proliferation and neovascularization	[121]
miRNA-200b-3p	Anti-angiogenic	Inhibits erythroblast transformation-related genes	[122]
HMGB1	Immune evasion	Expands regulatory B cells, facilitating tumor survival	[123]
Exosomal SMAD, Caveolin, MET, caveolins, S100, ITGαvβ5, OXL4, SDF-1α, IL-6, IL-8, AFP, and GGT	HCC progression	Cell adhesion, motility, invasive abilities, metastasis, angiogenesis, and cell proliferation	[124,125,126,127,128]
miRNA-92a-2-5p	Promotes metastasis	Enhances cancer cell invasion via AR/PHLPP/p-AKT/β-catenin	[129]
miRNA-125a/b	Impedes tumor-associated macrophages	Targets CD90 in tumor-associated macrophages	[130]

**Table 3 cancers-16-02552-t003:** ECVs in enhancing immunotherapy and managing therapy resistance.

Function	Example	Description	Context and Patient Use	Reference
Immunomodulation	PD-L1-enriched exosomes	Exosomes carrying PD-L1 inhibited T-cell activity, modulating the immune environment to favor tumor growth and impacting responses to immunotherapy	Oncology immunotherapy research—research phase	[174]
Drug Resistance	Exosomes carrying miRNA-1247-3p	Enhanced resistance to sorafenib in HCC by altering gene expression related to drug metabolism and cellular survival pathways	Chemoresistance mechanisms study—research phase	[135]
Chemotherapy Delivery	Exosomes for Doxorubicin delivery	Facilitated targeted delivery of Doxorubicin, enhancing drug specificity and minimizing off-target effects	Targeted chemotherapy research—early clinical trials	[178,179]
Gene Therapy	miRNA-220a/220b/429 mimics	Delivered miRNAs that regulate oncogenic pathways, providing a method for precision gene therapy	Gene therapy innovation—research phase	[174]
Chemoresistance Mechanisms	Exosomes carrying circRNA-SORE and miRNAs in chemoresistance	Studied circRNAs and miRNAs that enhance cellular mechanisms of resistance to chemotherapy agents	Molecular oncology exploration—research phase	[84,190,191,192,193,194,195]
Enhanced Immune Surveillance	Exosomes carrying HMGB1	Examined the role of HMGB1-bearing exosomes in modulating immune surveillance in HCC	Immune surveillance enhancement—research phase	[176]
Prognostic Biomarkers	Circulating exosomal PD-L1 levels	Developed and validated exosomal PD-L1 as a biomarker for assessing responses to immunotherapy	Biomarker development—early clinical trials	[174]
Therapeutic Drug Delivery	Exosomes in targeted drug delivery	Employed engineered exosomes for specific drug conveyance to tumor sites, reducing systemic toxicity	Drug delivery system development—early clinical trials	[175,176,177,178,179]
Circadian Influence on Therapy	Variability in exosome release by circadian rhythms	Circadian rhythms affect the secretion and composition of exosomes, influencing the response	Chronotherapy research—research phase	[8,200]

**Table 4 cancers-16-02552-t004:** Examples of circadian regulation of ECVs.

Example	Finding	Reference
Circadian Variation in ECV Quantity	ECV quantity extracted from peripheral blood, bone marrow, and lungs of mice exhibited time-dependent changes	[215]
Circadian Control of Exosomal Cargo	Flot1 regulated circadian control over MMP 14 in tendon fibroblast small ECVs	[200]
Impact of Night Shift Work on ECV Cargo	Night shift work disrupted circadian rhythms and exosomal cargo, influencing metabolic health	[216]
Circadian Variations in Plasma ECV Characteristics	Plasma ECVs were larger at 10:00 compared to 22:00 in HIV patients	[217]
Influence of Exosomal Cargo on Insulin Resistance	Exosomes from obese mice or patients with type II diabetes could induce insulin resistance in lean mice	[218,219,220]
Circadian Normalization Factor for Small ECV Biomarkers	Biomarker TSG101 levels in urine showed a circadian correlation with small ECV excretion in healthy rats	[221]

**Table 5 cancers-16-02552-t005:** Impact of circadian clock components on cancer progression via ECVs.

Impact of Circadian Clock	Description	Associated Conditions/Models	Reference
Circadian Component and Cancer Role			
BMAL1 Function	BMAL1 promotes metastasis in colorectal cancer through increased exosome secretion	Colorectal cancer The link between circadian control and tumor progression	[17]
SIRT1 Mechanisms	SIRT1 interacts with CLOCK-BMAL1 to modulate PER2 stability, affecting exosome secretion and tumor environment interactions	Facilitates tumor microenvironment degradation Breast cancer	[252,253,254,255]
SIRT1 Mechanisms	SIRT1 loss leads to enhanced exosome secretion, impacting breast cancer and diabetic nephropathy	Diabetic nephropathy	[256,257,258]
SIRT1 in Cancer	Shift-work-related miRNA-22-3p uptake by nurses is linked to increased insulin resistance, highlighting its potential as a biomarker for diabetes prevention	Ovarian cancer	[259]
Shift Work and its Effects			
Shift-Work-Induced Changes	Chronic shift work in mice alters intestinal flora and increases colonic permeability, affecting circadian gene expression via changes in plasma ECV components	Mouse model of chronic shift work	[251]
Night Shift Exosomal Impact	Night shift conditions lead to reduced insulin sensitivity in adipocytes through alterations in exosome content, affecting core clock genes	Simulated shift work study Affects core clock genes and metabolic functions	[216]
Shift Work and Diabetes Risk	Shift-work-related miRNA-22-3p uptake by nurses is linked to increased insulin resistance, highlighting its potential as a biomarker for diabetes prevention	Shift nurses	[267]
Exosomal miRNAs and Circadian Genes	miRNA-3614-5p as a messenger of circadian misalignment in night shift workers	Contributes to metabolic dysfunction	[216]
Circadian Genes and miRNA	Circulating miRNAs like miRNA-219, miRNA-152, miRNA-494, and miRNA-142-3p regulate clock genes	Affects BMAL1 and PER1 in circadian regulation	[268,269,270,271,272]
Circadian Genes and miRNA	Exosomal miRNAs modulate peripheral circadian oscillators in various disease models	Has impact on glioma progression and Parkinson’s disease	[9,264]
Melatonin’s Therapeutic Role			
Melatonin-Enhanced Exosomes	Melatonin pre-treatment enhances anticancer and anti-inflammatory properties of exosomes, improving their therapeutic effectiveness in various clinical settings	Cancer therapies, various models	[273,274,275]
Anti-Inflammatory Effects	Melatonin-treated exosomes significantly reduce inflammatory markers and PD-L1 expression in macrophages, aiding in cancer therapy and reducing immune evasion	Hepatocellular carcinoma	[247]

## Data Availability

No new data were created or analyzed in this study.

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
