# Peer review of "Extracellular Vesicles, Circadian Rhythms, and Cancer: A Comprehensive Review with Emphasis on Hepatocellular Carcinoma"

_cancers, 2024, doi:10.3390/cancers16142552_

Round 1

Reviewer 1 Report

Comments and Suggestions for Authors

This manuscript titled as “Extracellular vesicles, Circadian Rhythms, and Cancer: A Comprehensive Review with Emphasis on Hepatocellular Carcinoma”, provides a review of the interplay between ECVs and circadian rhythms in the context of HCC. It synthesizes current knowledge and emphasizes the importance of understanding circadian influences on exosome biology to improve cancer diagnostics and treatments. The manuscript presents relevant findings that might provide new sights for both researchers and clinicians in the field of oncology. Novelty can be observed in the title of this manuscript. However, the substantial content is absent. More in-depth content is required for this article. In addition, apparent flaws can be also observed in this manuscript. Thus, major revision is recommended. 

1.This manuscript discusses the “ECVs in Immunotherapy and Therapy Resistance”, but authors only summarized the relationship of ECV and PD-1/PD-L1 inhibition therapy. How the effects of ECV on other types of  ICIs? And how about other types of immunotherapy (CART? Cancer vaccines? etc)

2.A flowchart or diagram summarizing the biogenesis pathways of exosomes for better visualization is recommended. 

3.Authors should provide more specific mechanisms by which exosomes influence the tumor microenvironment.

4.Authors should provide a comparison with traditional biomarkers and exosomal biomarkers, and discuss the current limitations and challenges in the clinical implementation of exosomal biomarkers.

5. How can ECVs be specifically applied to the treatment or prediction of HCC based on the content summarized in this article? What are the impacts, advantages, and disadvantages compared to current clinical methods? What are the potential challenges and more specific future research directions? What is the current level of research and clinical application?

Minor Comments:

6.Please check for consistency in terminology (e.g., use either "extracellular vesicles" or "ECVs" uniformly).

7.Some references appear multiple times; ensure they are consolidated and check for any missing citations.

Author Response

Dear Reviewer,

Thank you for allowing us to submit a revised draft of the manuscript titled “Extracellular Vesicles, Circadian Rhythms, and Cancer: A Comprehensive Review with Emphasis on Hepatocellular Carcinoma” to Cancers. We appreciate the time and effort that you have dedicated to providing valuable feedback. The insightful comments have clearly improved our manuscript. Changes have been highlighted in the “tracked changes” version of the manuscript.

Below is a point-by-point response to the reviewers’ comments and concerns:

Comments from Reviewer 1

Comments and Suggestions for Authors

This manuscript, titled “Extracellular vesicles, Circadian Rhythms, and Cancer: A Comprehensive Review with Emphasis on Hepatocellular Carcinoma”, provides a review of the interplay between ECVs and circadian rhythms in the context of HCC. It synthesizes current knowledge and emphasizes the importance of understanding circadian influences on exosome biology to improve cancer diagnostics and treatments. The manuscript presents relevant findings that might provide new sights for both researchers and clinicians in the field of oncology. Novelty can be observed in the title of this manuscript. However, the substantial content is absent. More in-depth content is required for this article. In addition, apparent flaws can be also observed in this manuscript. Thus, major revision is recommended. 

  1. This manuscript discusses the “ECVs in Immunotherapy and Therapy Resistance”, but authors only summarized the relationship of ECV and PD-1/PD-L1 inhibition therapy. How the effects of ECV on other types of ICIs? And how about other types of immunotherapy (CART? Cancer vaccines? etc)

Thank you for your valuable feedback. Following your advice, we added new paragraphs to section 3. 5. ECVs in Immunotherapy and Therapy Resistance to discuss the effects of other ICIs and additional immunotherapies, such as CART-cell therapy and cancer vaccines. (338-365) and (377-383)

  1. A flowchart or diagram summarizing the biogenesis pathways of exosomes for better visualization is recommended.

Thank you for your comment. We included more details about the ESCRT-dependent and ESCRT-independent pathways in the text (Section 2; lines 113-120) and modified Figure 1. See the Figure below:

  1. Authors should provide more specific mechanisms by which exosomes influence the tumor microenvironment.

We appreciate your insightful feedback. We have incorporated it into a new section (Section 3.3. ECVs and Tumor Microenvironment; lines 218-246).

  1. Authors should provide a comparison with traditional biomarkers and exosomal biomarkers, and discuss the current limitations and challenges in the clinical implementation of exosomal biomarkers.

Thank you for your comment; following your considerate suggestion, we included a discussion comparing traditional and exosomal biomarkers, as well as an analysis of the current limitations and challenges in the clinical implementation of exosomal biomarkers, to the conclusion (section 5; lines 542-555). 

  1. How can ECVs be specifically applied to the treatment or prediction of HCC based on the content summarized in this article? What are the impacts, advantages, and disadvantages compared to current clinical methods? What are the potential challenges and more specific future research directions? What is the current level of research and clinical application?

Thank you for your thoughtful comment. We addressed this comment by discussing the current limitations and challenges of implementing exosomal biomarkers clinically (Section 5; lines 591-599).

6.Please check for consistency in terminology (e.g., use either "extracellular vesicles" or "ECVs" uniformly).

Thank you for pointing this out. We unified the consistency for the abbreviations:

  •         Extracellular vesicles (ECVs)
  •         Extracellular vesicle (ECV) 

We also unified consistency for miRNAs abbreviations.

7.Some references appear multiple times; ensure they are consolidated and check for any missing citations.

Thanks for your comment. We checked and deleted the ones that were duplicated.

Reviewer 2 Report

Comments and Suggestions for Authors

The authors offer a detailed overview of the complex interaction between extracellular vesicles (ECVs)/exosomes and circadian rhythms, with an emphasis on the role of this interaction in hepatocellular carcinoma (HCC). An innovative emphasis is placed on the controlled release of exosomes and their potential as biomarkers for the early detection of cancer and metastases. Potential clinical applications of ECVs are also reviewed, particularly their use as diagnostic tools and drug delivery vehicles.

Minor comments:

1.....Abstract: his.... to be changed to This>;to add a graphical abstract.

2. Introduction -In the text is used once ECVs , second ECV ? to unify!

3. Figure 1 is well expressed and described.

4. Rows 115-121 to unfold

5. Row 150-158 to be revised

6. Figure 2 is well expressed and described.

7. Row 188- 195 to be revised; too much abbreviation; difficult readability.

8. Figure 3 is well expressed and described.

9. The lipids' role is not well described. 

10. Figure 4 is well expressed and described.

11. Row 323 - 337 to be revised; too much abbreviation; difficult readability.

12. In section 5. Additional studies of exosomal cargo sorting mechanisms to be listed to enhance the contribution of the manuscript;

Innovative techniques for non-invasive ...............:

a) isolation of cell-specific exosomes,

b).......

(c) promising avenues for early disease detection and intervention.

..... the use of ASGR1 to purify hepatocyte-derived exosomes

13. Future objectives/prospects

Comments on the Quality of English Language

Minor editing of English language required

Author Response

Dear Reviewer,

Thank you for allowing us to submit a revised draft of the manuscript titled “Extracellular Vesicles, Circadian Rhythms, and Cancer: A Comprehensive Review with Emphasis on Hepatocellular Carcinoma” to Cancers. We appreciate the time and effort that you have dedicated to providing valuable feedback. The insightful comments have clearly improved our manuscript. Changes have been highlighted in the “tracked changes” version of the manuscript.

Below is a point-by-point response to the reviewers’ comments and concerns:

Comments from Reviewer 2

The authors offer a detailed overview of the complex interaction between extracellular vesicles (ECVs)/exosomes and circadian rhythms, with an emphasis on the role of this interaction in hepatocellular carcinoma (HCC). An innovative emphasis is placed on the controlled release of exosomes and their potential as biomarkers for the early detection of cancer and metastases. Potential clinical applications of ECVs are also reviewed, particularly their use as diagnostic tools and drug delivery vehicles.

Minor comments:

  1. Abstract: his.... to be changed to This>;to add a graphical abstract.

Thank you so much for pointing this out. We corrected the text.

  1. Introduction -In the text is used once ECVs, second ECV? to unify!

Thank you for pointing this out. We unified the consistency for the abbreviations:

  •         Extracellular vesicles (ECVs)
  •         Extracellular vesicle (ECV) 

We also unified consistency for miRNAs abbreviations.

  1. Figure 1 is well expressed and described.

Thank you for your kind comment. Although there were no specific remarks about Figure 1, we have revised it to incorporate both ESCRT-dependent and -independent pathways, following suggestions from another reviewer.

  1. Rows 115-121 to unfold

Thank you for your comment; we included more details about the ESCRT-dependent and ESCRT-independent pathways in section 2, lines 113-120, and Figure 1. Please see the Figure below.

  1. Row 150-158 to be revised

We modified this paragraph to improve readability. Please check Section 3, Lines 152-155, for further details. 

  1. Figure 2 is well expressed and described.

Thank you for your kind comment. 

  1. Row 188- 195 to be revised; too much abbreviation; difficult readability.

We improved the readability. See Section 3.1, Lines 188-196, for further details.

  1. Figure 3 is well expressed and described.

Thank you for your kind comment. 

  1. The lipids' role is not well described. 

Thank you for your thoughtful comments. This review primarily focuses on ncRNA and protein biomarkers, and due to the extensive nature of lipidomics, discussing it in detail here goes beyond the scope of the manuscript. However, in the section “3.4.  Diagnostic Biomarkers in HCC” (Lines 307 to 310), we have included a clinical study where we mentioned some lipid biomarkers.

  1. Figure 4 is well expressed and described.

Thank you for your kind comment. 

  1. Row 323 - 337 to be revised; too much abbreviation; difficult readability.

We improved the readability. Please see Section 4, lines 397- 410 for further details.

  1. In section 5. Additional studies of exosomal cargo sorting mechanisms to be listed to enhance the contribution of the manuscript;

Innovative techniques for non-invasive ...............:

  1. a) isolation of cell-specific exosomes,

b).......

(c) promising avenues for early disease detection and intervention.

..... the use of ASGR1 to purify hepatocyte-derived exosomes

Following your valuable suggestion, we expanded Section 5 to include additional studies on exosomal cargo sorting mechanisms. We've also discussed innovative techniques for the non-invasive isolation of cell-specific exosomes and promising avenues for early disease detection and intervention. Currently, research on the specific isolation of liver cell-derived exosomes from blood is limited. Therefore, we focused on the use of ASGR1 in the manuscript. Please see Section 5, lines (530-535) and (542-555) for further details.

  1. Future objectives/prospects

Thanks for your valuable advice, following your suggestion, we added more detailed objectives and prospects to the conclusion (Section 5, lines 591-599).

Round 2

Reviewer 1 Report

Comments and Suggestions for Authors

The review provides a comprehensive overview of the interplay between extracellular vesicles (ECVs), particularly exosomes, and circadian rhythms in the context of hepatocellular carcinoma (HCC). It synthesizes existing knowledge on various roles of ECVs in cancer, highlighting their importance in intercellular communication, immune modulation, drug resistance, and angiogenesis. This manuscript also points out the potential clinical applications of ECVs, particularly their use as diagnostic tools and drug delivery vehicles, providing certain clinical significance.